# UFMC-Based Underwater Voice Transmission Scheme with LDPC Codes

**Chin-Feng Lin**

Department of Electrical Engineering, National Taiwan Ocean University, Keelung City 202301, Taiwan; lcf1024@mail.ntou.edu.tw

**Abstract:** An underwater universal filtered multicarrier (UFMC)-based voice transmission scheme is proposed using a 512-point inverse discrete Fourier transform, utilizing 10 sub-bands, and that each had 20 subcarriers. In this proposed UFMC method, the adaptive modulation technologies with 4 quadrature amplitude modulation (QAM), 16-QAM, and low-density parity-check (LDPC) channel coding were integrated. Additionally, the bit error rate (BER), transmission power weighting, the ratios of power-saving, and underwater voice transmission performance with perfect channel estimation (PCE), and 5% and 10% channel estimation errors (CEEs) were investigated. The underwater voice transmission had a BER quality of service $10^{-3}$. Simulation results showed that the PCE outperformed 5% and 10% CEEs, under 4-QAM, with gains of 0.5 and 0.9 dB, respectively, and a BER of $4 \times 10^{-4}$. The PCE outperformed 5% and 10% CEEs, under 16-QAM, with gains of 0.5 and 2.4 dB, respectively, and a BER of $8.5 \times 10^{-4}$. The proposed UFMC scheme can be applied to underwater voice transmission with a BER below $10^{-3}$ The proposed system showed a superior capability to contend with additive white Gaussian noise, underwater multipath channel fading, and CEEs.

**Keywords:** universal filtered multicarrier (UFMC); underwater; voice; low-density parity-check (LDPC)





## 1. Introduction

Underwater acoustic communication (UAC) is an interesting research area with many useful applications. At present, UAC channels in shallow water are characterized by band-limitation, extensive inter-symbol interference, and a large Doppler spread. Shallow water is defined as water with a depth between 10 and 200 m. For a water depth of 20 m, the transmission range of 50 km, source depth of 19 m, and a receiver depth of 17.2 m, the multipath spread is 5 ms when using binary phase-shift keying (BPSK) modulation with a 17 kHz carrier frequency, and a 4 kHz bandwidth [1]. For UAC, higher frequencies lead to shorter communication ranges. Zhou et al. [2] investigated UAC characteristics using quadrature phase-shift keying (QPSK) modulation with a data rate of 27.2 kb/s, the transmission range of 1500 m, the transmission bandwidth of 21.25 kHz, and a carrier frequency of 85 kHz in a shallow water region. Indeed, robust and reliable underwater wireless networks are rapidly expanding to explore the undersea world [3]. Cheng et al. [3] demonstrated the concept of cooperative orthogonal frequency-division multiplexing (OFDM) UAC, underwater acoustic channel modeling, and an adaptive system design, with optimal power allocation and distribution, for short-range UAC.

OFDM has been applied in standards for the digital subscriber line (DSL), Institute of Electrical and Electronics Engineers (IEEE) 802.11n, IEEE 802.16, third-generation partnership project long-term evolution (3GPP-LTE), and LTE advanced standards. Filter bank multicarrier (FBMC), generalized frequency division multiplexing (GFDM), and universal filtered multicarrier (UFMC) are derived from OFDM principles [4]. FBMC adopts a subcarrier-based filter operation to achieve lower out-of-band leakage, whereas GFDM uses one cyclic prefix (CP) for many OFDM symbols to improve the bandwidth efficiency of data transmission. Cai et al. [5] investigated several orthogonal modulation approaches, such as

FBMC, GFDM, UFMC, and filtered OFDM for fifth-generation (5G) networks. Compared with traditional OFDM, advanced modulation technologies have high spectral efficiency, loose synchronization requirements, and flexibility to support transceivers with various transmission data rates. However, FBMC and GFDM transceivers using subcarrier-based filtering are more complex. For the UFMC scenario, subband-based filtering was utilized. Schaich et al. [6] expounded on the waveform design principles of UFMC and FBMC, which they reported as suitable for user-centric transmission.

The high spectral efficiency of UFMC, a new data transmission method, makes it ideal for use in future 5G mobile networks. Bochechka et al. [7] designed a UFMC system characterized by 80 subbands with 12 subcarriers each, 16 quadrature amplitude modulation (QAM), an inverse discrete Fourier transform (IDFT) with a size of 1024, and a Dolph–Chebyshev filter; they also demonstrated bit error rate (BER) performance in an additive white Gaussian noise (AWGN) channel. UFMC adopts the approach of filtering a block of subcarriers (subband) to reduce the transmission requirement of additional training signals and synchronization of complex systems; it can be applied to the Internet of things (IoT)/machine-to-machine (M2M) device communication. Kumar et al. [8] found that the symbol error rate for UFMC with QPSK modulation is better than that of OFDM with QPSK modulation at the same frequency offset.

To overcome multipath fading channel effects in UFMC technology, researchers have considered foregoing the CP for QPSK modulation, 16-QAM, and 64-QAM [9]. UFMC is a novel multicarrier technique that has the advantages of OFDM while also avoiding its drawbacks, such as strict time and frequency synchronization and high out-of-band emissions [10]. Compared with OFDM, UFMC has better performance in addressing weaknesses using sub-band filters. Wang et al. [11] integrated active interference cancellation (AIC) into a UFMC system to enable further reduction of inter-sub-band interference and more reliable communication. Rani et al. [12] described aspects of UFMC technology and explored the PAPR and BER performance of UFMC technology for various modulation schemes. Simulation results have shown that BER values for UFMC increase with an increase in the number of bits per subcarrier and that the 16-QAM method is suitable for UFMC technology. Further, the spectral efficiency of UFMC is lower than that of FBMC.

An intermediate scheme between OFDM and FBMC, UFMC combines the simplicity of OFDM and the anti-interference performance of FBMC. UFMC schemes use a Dolph–Chebyshev filter with a finite impulse response to shape the waveform and enhance anti-interference performance [13]. Wen et al. [13] proposed an effective design of a waveform-shaping filter to eliminate out-of-band emission, suppress spectral side-lobe levels, and employ well-designed anti-interference filters for symbol shaping. Their proposed method considers the design parameters of passband ripple, stopband attenuation, transition width, and Nyquist condition, and hence can produce better BER outcomes. The filtering operation in UFMC is adopted on a group of consecutive subcarriers (sub-band shaping); the filter length and implementation complexity are decreased significantly compared with FBMC. Aspects of the spectral efficiency, numerical complexity, and power amplifier nonlinearity for UFMC, FBMC, GFDM and resource-block filtered OFDM have been demonstrated [14]. These new transmission schemes have OFDM advantages while also overcoming their drawbacks. Compared with OFDM, FBMC has the best spectral containment, GFDM has the smallest complexity overhead, and UFMC has the best compatibility.

Zhang et al. [15] proposed a CP-based UFMC system to achieve interference-free transmission and derived an analytical model with the desired signal, intersymbol interference, intercarrier interference, and noise-level. Chen et al. [16] formulated a novel adaptive filter configuration algorithm by adaptively designing the parameters of finite impulse response filters. Their adaptive filter configuration algorithm can efficiently combat different carrier-frequency offsets that are caused by the interference of multiple users. Adaptive modulation and power allocation for each subcarrier are integrated to achieve better BER performance and a higher transmission data rate compared with the traditional UFMC-based transmission system. Another approach to achieving excellent BER performance in

multicarrier UAC is the use of nonbinary low-density parity-check (LDPC) channel coding, which was investigated by Huang et al. [17]. Liu et al. [18] reported on an LDPC decoder and a decision feedback equalizer that iteratively exchanges soft information. Their simulation results showed that a BER performance $10^{-5}$ in the proposed LDPC decoder increases by 0.8 dB compared with turbo channel coding with the same coding rate. Amini et al. [19] developed a robust FBMC method in doubly dispersive UAC channels.

In the area of underwater acoustic multimedia communication, Lin et al. [20–22] investigated transmission schemes that incorporate direct-mapping orthogonal variable spreading factor (OVSF), multi-input multi-output (MIMO)-OFDM, or MIMO Gold sequence (GS) OVSF/OFDM. Lin et al. [23] proposed an FBMC-based underwater transmission scheme for voice and image signals and integrated LDPC channel coding, adaptive BPSK or offset QAM (OQAM), and a power assignment mechanism into an underwater voice and image transmission system. In a later work, Lin et al. [24] designed a direct mapping (DM)-based underwater transmission scheme for voice and image signals that integrated the FBMC transmission method. Meanwhile, 5GNOW_D3.1_v1.1 [25] demonstrated a UFMC-based transceiver architecture.

The remainder of this paper is structured as follows: Section 2 provides the design methods of the proposed UFMC underwater voice transmission. Section 3 presents the simulation results using the proposed UFMC-based underwater transceiver. Our conclusions are presented in Section 4.

## 2. System Models

An UFMC-based voice transceiver design with an LDPC code is an interesting research area. In this section, an overview of the proposed system model is presented. Figure 1 depicts the proposed UFMC-based underwater transceiver with an LDPC code for voice signals, whereas Table 1 lists its parameters. UFMC modulation with a (2000, 1000) LDPC code was an advanced UAC scheme, and high spectral efficiency, high transmission data rate, low transmission BER, and real-time voice transmission could be achieved. The underwater channel bandwidth was 20 kHz, and 512-point inverse discrete Fourier transform (IDFT), utilizing 10 sub-bands with 20 subcarriers each, was used. G.729 encoder can compress voice signal, and voice signal transmission bits were reduced. The quality of service of BER for underwater voice signal transmission was below $10^{-3}$. The adaptive 4-QAM and 16-QAM modulations with the integration of 1/30, 2/30, . . . , 30/30 power levels mechanism, 60 transmission mode was achieved to combat diversification underwater channel fading. The quality of service of BER for underwater voice signal transmission was achieved, real-time and low-power underwater voice signal transmissions were also achieved.

First, a voice signal was used as the input of a G.729 encoder, and then extracted a voice G.729 bitstream as the output. This voice G.729 bitstream was then used as the input of a (2000, 1000) LDPC encoder, and a voice LDPC bitstream was extracted as the output. A voice LDPC bitstream was applied to an adaptive modulation scheme with 4-QAM, and 16-QAM, with adaptive QAM symbols were extracted as the output. The proposed scheme adopted carrier sense multiple access with collision avoidance (CSMA/CA) for multiuser communication. The UFMC solution was integrated into the proposed underwater voice transmission technology. The baseband adaptive UFMC transmitted signal for the $k$th user is expressed as follows:

$$Z_k = p_k X_k$$
$$X_k = \sum_{i=1}^{B} F_{ik} V_{ik} U_{ik} \tag{1}$$

where $p_k$ is the transmission power weighting of the $k$th user. $U_{ik}$ is a vector of the complex adaptive QAM symbols for the $i$th sub-band, $k$th user, and the number of subcarriers for the $i$th sub-band, and the $k$th user, transformed into the IDFT matrix $V_{ik}$. $V_{ik}$ has $N \times n_{ik}$ dimensions composed of the relevant columns of the inverse Fourier matrix, arranged according to the respective subband position within the overall available frequency range. $F_{ik}$ is a Toeplitz matrix with a Chebyshev filter impulse response. The dimension of $F_{ik}$ is

$(N + N_{filter} - 1) \times N$. $B$, $N$, and $N_{filter}$ are the number of sub-bands, the overall number of subcarriers, and length of the FIR filter coefficients, respectively.

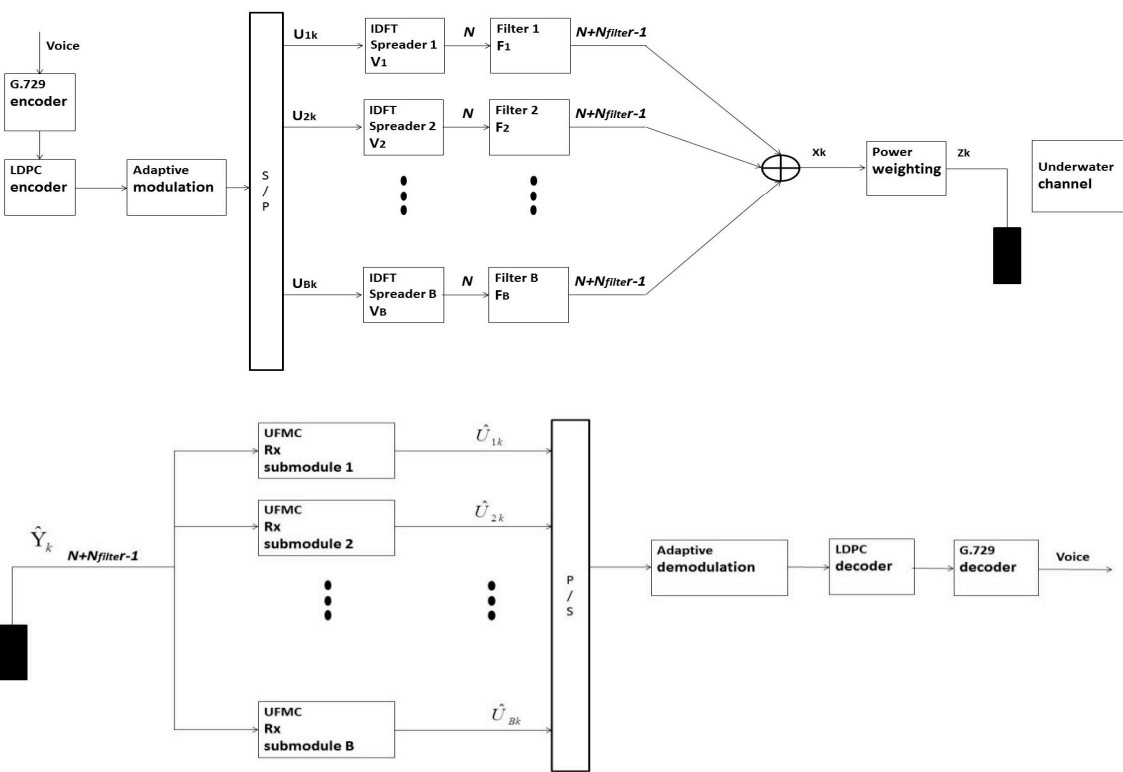

**Figure 1.** The proposed universal filtered multicarrier (UFMC)-based underwater transceiver with a low-density parity-check (LDPC) code for voice signals.

**Table 1.** Simulation parameters of the proposed UFMC-based underwater transceiver with an LDPC code.

| Technology | Technology Characteristics |
|---|---|
| UFMC modulation | 5GNOW_D3.1_v1.1 [25] |
| Number of IDFT points | 512-points |
| Number of subbands | 10 |
| Subband size | 20 |
| Filter method | Chebyshev filtering operation |
| Channel bandwidth | 20 kHz |
| Adaptive modulation | 4-QAM, and 16-QAM |
| Channel coding | (2000, 1000) LDPC code encoder with a code rate of 1/2, a column weight of 3, a row weight of 6 |
| Voice compression method | G.729 encoder |
| Power levels | 1/30, 2/30, . . . , 30/30 |
| BER limits for voice transmission | $10^{-3}$ |

The signal can be rewritten as follows:

$$
\begin{aligned}
\overline{F}_K &= [F_{1k}, F_{2k}, \cdots, F_{nk}] \\
\overline{V}_k &= diag(V_{1k}, V_{2k}, \cdots, V_{Bk}) \\
\overline{U}_k &= [U_{1k}^T, U_{2k}^T, \cdots, U_{Bk}^T]^T \\
Z_k &= s_k \overline{F}_k \overline{V}_k \overline{U}_k
\end{aligned}
\tag{2}
$$

The received signal vector after propagation through the underwater channel for the $k$th user is expressed as follows:

$$Y_k = H_k Z_k + N_0 \tag{3}$$

where $N_0$ is additive white Gaussian noise (AWGN) for the $k$th user, and the convolution matrix $H_k$ with Toeplitz structure for the $k$th user, is generated by the time-domain underwater channel impulse response. The received signal vector with a perfect channel estimation (PCE) and channel estimation errors (CEEs) for the $k$th user is expressed as follows:

$$\hat{Y}_k = Y_K (\hat{H}_k)^{\mathrm{T}} \tag{4}$$

The convolution matrix $\hat{H}_k$ is generated by the time-domain underwater channel impulse response with PCE and CEEs. The time-domain underwater channel impulse response with a PCE is equal to the time-domain underwater channel impulse response. The output signal vector after the concatenation of inverse filtering and discrete Fourier transform for the $k$th user is expressed as follows:

$$\hat{U}_k = \hat{Y}_k (\overline{FV})^+ \tag{5}$$

$A^+$ is the Moore–Penrose-inverse of a matrix. In the proposed UFMC-based underwater voice solution with an LDPC code, a UFMC demodulation solution, adaptive QAM demodulation, LDPC decoder, and G.729 decoder were integrated. This scheme utilized 512 IDFT, 10 sub-bands with 20 subcarriers each, a channel bandwidth of 20 kHz, and a Chebyshev filtering operation. The power weighting factors were $1/30, 2/30, \ldots, 30/30$. A power assignment mechanism was also integrated to achieve a lower transmission power consumption or higher transmission data rates compared with systems without a power assignment mechanism. The power assignment mechanism for the proposed UFMC-based underwater acoustic voice transmission scheme with an LDPC code is summarized as follows:

Step 1: Select the 16-QAM modulation mode;

Step 2: Assign the initial value of $p_k$ as 15/30 for voice packets;

Step 3: Measure the received *SNR* for voice packets;

Step 4: If the measured *SNR* of the received signal exceeds the threshold *SNR* at which the required BER for voice packets is achieved, then update $p_k$ as $p_k = p_k - \Delta$. $\Delta$ is $\frac{1}{30}$. If $s_k > \frac{1}{30}$, go to step 3; otherwise, proceed to step 6.

Step 5: If the measured *SNR* of the received signal is less than the threshold SNR at which the required BER for voice packets is achieved, then update $p_k$ as $p_k = p_k + \Delta$. $\Delta$ is $\frac{1}{30}$. If $s_k < 1$, go to step 3; otherwise, proceed to step 7.

Step 6: If the modulation mode is not 16-QAM, change the modulation mode to16 QAM, and go to step 3; otherwise, proceed to step 3;

Step 7: If the modulation mode is not 4-QAM, change the modulation mode to 4 QAM, and go to step 3; otherwise, proceed to step 3;

In the proposed power assignment mechanism $\Delta$ was to increase or decrease transmission power weighting for each power assignment loop. Higher transmission power weighting could contend with higher channel fading, and lower transmission power weighting could contend with lower channel fading.

## 3. Simulation Results

Figure 2 depicts the BER performances of the UFMC-based underwater transceiver with an LDPC code, PCE and CEEs of 5% and 10%. The circle, square, and triangle symbols denote the BER performances with PCE and CEEs of 5% and 10%, respectively, in Figures 2–4. The solid and dashed lines denote the BER performances using 4-QAM, and 16-QAM, respectively, in Figures 2–4. The Matlab-based underwater channel model [26] was adopted. The parameters of the adopted MATLAB-based underwater channel model are listed in Table 2. The transmitter was set 3 m beneath the sea surface, the receiver 2 m beneath the sea surface, the water 14.5 m deep, the direct path at a range of 100 m, the

carrier frequency at 40 kHz, and the channel bandwidth at 20 kHz. In the underwater channel model, the water sound speed was 1539 m/s. In addition, the channel delay profile was generated by using a MATLAB-based underwater channel model [26].

The BER of the proposed system was below $10^{-3}$, thereby achieving the quality of services requirements for underwater voice transmissions. The simulation results showed that as the SNR increased, the BERs for the proposed UFMC-based underwater transceiver decreased. As the CEEs increased, the BERs for the proposed UFMC-based underwater transceiver increased. The SNRs using 4-QAM with PCE and CEEs of 5%.

**Table 2.** Parameters of the adopted underwater channel model [26].

| Parameter | Value |
|---|---|
| Depth of transmitter beneath the sea surface, meters | 3 |
| Depth of receiver beneath the sea surface, meters | 2 |
| Water depth, meters | 14.5 |
| Water sound speed | 1539 m/s |
| Range of direct path, meters | 100 |
| Carrier frequency | 40 kHz |
| Channel bandwidth | 20 kHz |

And 10% were 14.0671, 14.5223, and 14.9894 dB, respectively, for the proposed system with a BER of $4 \times 10^{-4}$. As the CEEs increased, the SNRs increased, at the same transmission BER of $4 \times 10^{-4}$. The SNRs using 16-QAM with PCE and CEEs of 5% and 10% were 21.0568, 21.5490, and 23.4679 dB, respectively, for the proposed system with a BER of $8.5324 \times 10^{-4}$. The SNRs using 16-QAM were larger than those when using 4-QAM, at the same transmission BER and CEE. The (2000, 1000) LDPC code showed superior error-correcting capability in the proposed UFMC-based underwater transceiver in the BER range $10^{-2}$ to $4 \times 10^{-4}$. The 4-QAM likewise had a superior capability to overcome channel fading, whereas 16-QAM, a high-order modulation method, could achieve high data transmission rates.

Figure 3 shows the transmission power weighting of the UFMC-based underwater transceiver with an LDPC code and a BER of $4 \times 10^{-4}$, for PCE and CEEs of 5% and 10%, respectively. The AWGNs with spectral densities (*No*) were 0.0274, 0.0247, 0.0222, 0.00548, 0.0049, and 0.0031, for 4-QAM with a PCE, 4-QAM with a CEE of 5%, 4-QAM with a CEE of 10%, 16-QAM with a PCE, 16-QAM with a CEE of 5%, and 16-QAM with a CEE of 10%, respectively, at a transmission power weighting of 21/30. The PCE outperformed CEEs of 5% and 10%, under 4-QAM, with *No* gains of 0.0027 and 0.0052, respectively, at a transmission power weighting of 21/30. Although the CEE increased, No decreased. The 4-QAM outperformed 16-QAM, under CEEs of 10%, with *No* gains of 0.0243, at a transmission power weighting of 21/30. The *No* of 4-QAM was larger than that of 16-QAM under the assumption of the same transmission power weighting and CEE. Figure 4 shows the power saving ratios of the UFMC-based underwater transceiver with an LDPC code and a BER of $4 \times 10^{-4}$ for PCE and CEEs of 5% and 10%.

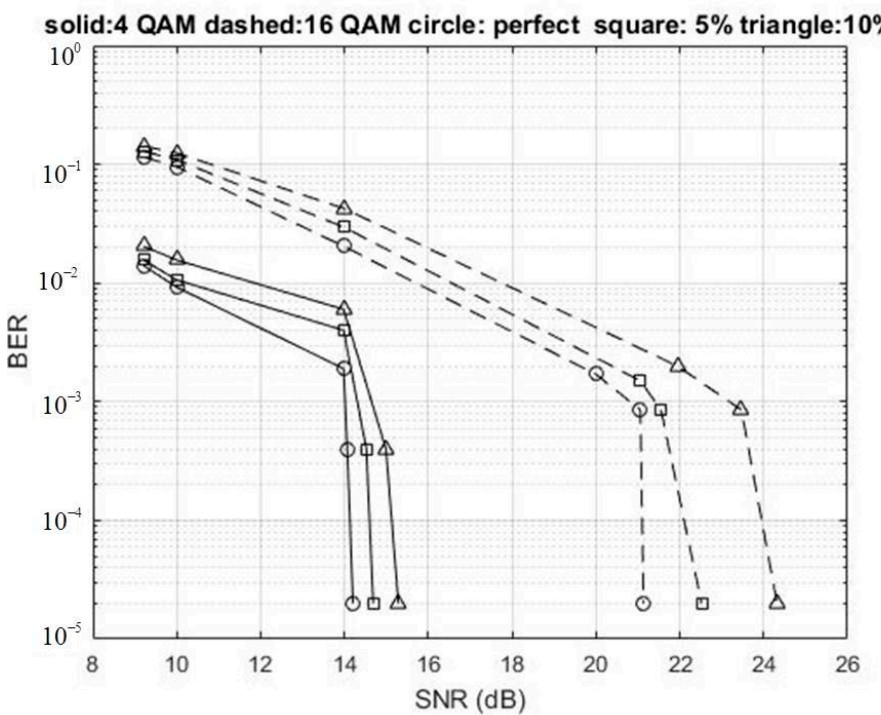

**Figure 2.** Quadrature amplitude modulation (QAM) bit error rate (BER) performances of the UFMC-based underwater transceiver with an LDPC code, a perfect channel estimation (PCE) and channel estimation errors (CEEs) of 5% and 10%.

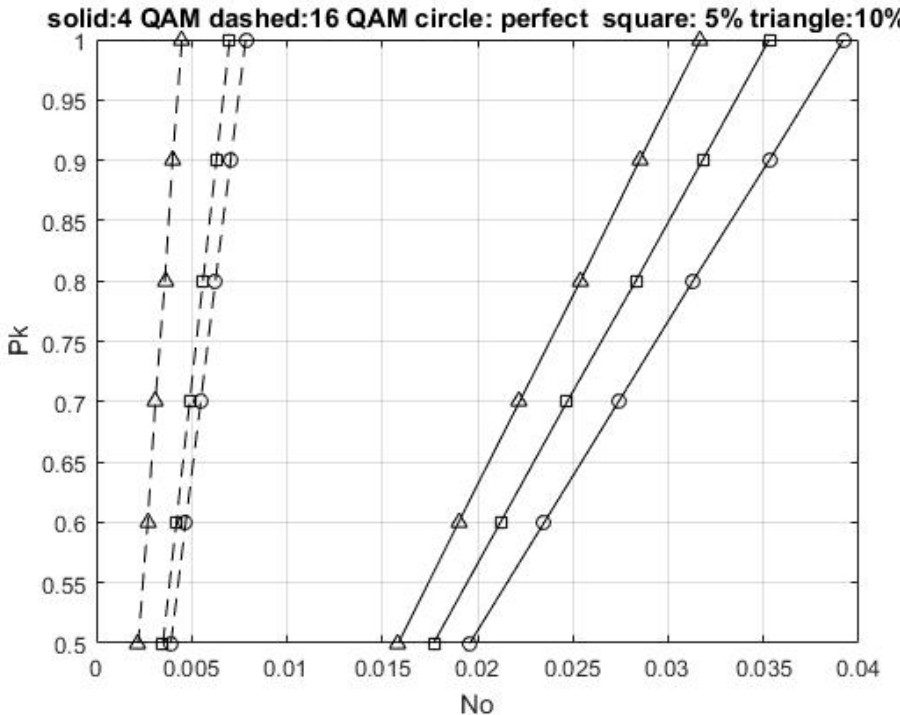

**Figure 3.** Transmission power weighting of the UFMC-based underwater transceiver with an LDPC code and a BER of $4 \times 10^{-4}$ for PCE and CEEs of 5% and 10%.

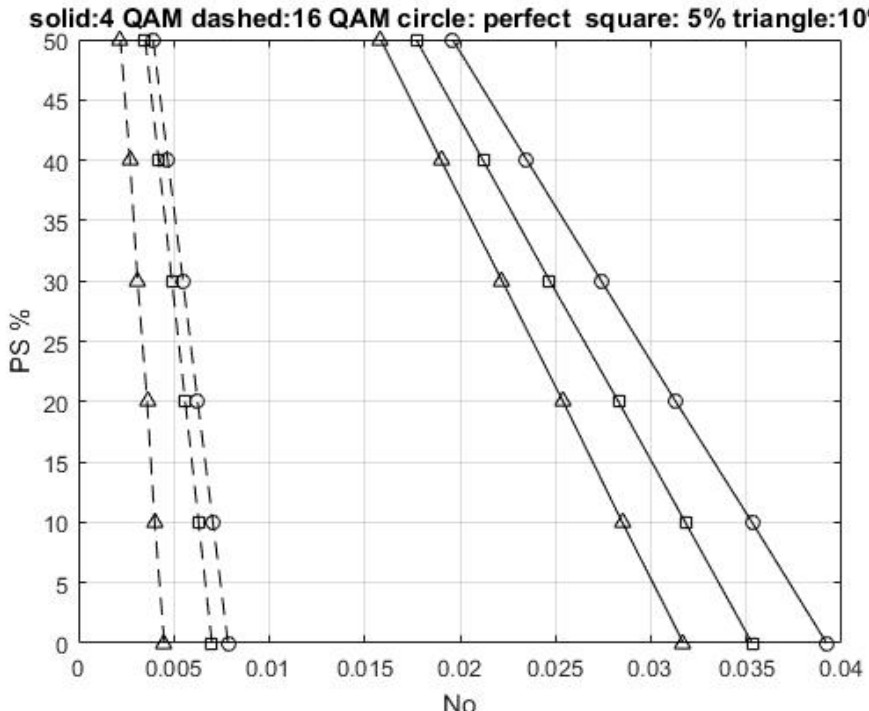

**Figure 4.** Power saving ratio of the UFMC-based underwater transceiver with an LDPC code and a BER of $4 \times 10^{-4}$ for PCE and CEEs of 5% and 10%.

The power-saving ratio of the UFMC-based underwater transceiver is defined as follows:

$$PS = (1 - p_k) \times 100\% \tag{6}$$

where $p_k$ is the transmission power weighting of the $k$th user.

For 4-QAM, the power saving ratios for the *No* of 0.0392, 0.0353, 0.0313, 0.0274, 0.0235, and 0.0196, were 0%, 10%, 20%, 30%, 40%, and 50%, respectively, in the proposed system with a BER of $4 \times 10^{-4}$, for PCE. The power-saving ratios for the *No* of 0.0317, 0.0285, 0.0254, 0.0222, 0.0190, and 0.0158, were 0%, 10%, 20%, 30%, 40%, and 50%, respectively, in the proposed system with a BER of $4 \times 10^{-4}$, for CEE of 10%. The PCE outperformed the CEES of 10%, under 4-QAM, with *No* gains of 0.0038, at a power-saving ratio of 50%. As the CEE increased, *No* decreased.

For 16-QAM, the power saving ratios for the *No* of 0.0045, 0.0040, 0.0036, 0.0031, 0.0027, and 0.0022, were 0%, 10%, 20%, 30%, 40%, and 50%, respectively, at the proposed system with a BER of $8.5 \times 10^{-4}$, for CEE of 10%. The 4-QAM outperformed 16-QAM, under a CEEs of 10%, with *No* gains of 0.0136, at a power-saving ratio of 50%. The No of 16-QAM was lower than that of 4-QAM.

The received voice signals used 16-QAM with a BER of $8.5 \times 10^{-4}$, for CEEs of 5%. The mean square error (MSE) between the original and received voice signals was 0.000015. The MSE of the original and received voice signals is expressed as follows:

$$MSE = \frac{1}{T} \sum_{i=1}^{T} \left( O_i - \hat{O}_i \right)^2 \tag{7}$$

where $O_i$ is the original voice signal, $\hat{O}_i$ is the received voice signal, and $T$ is the length of the original voice signal. The received voice signal was clear at the received transmission BER below $10^{-3}$. The MSEs between the original and the received voice signals using 4-QAM with a BER of $4 \times 10^{-4}$ for PCE and CEEs of 5% and 10%, were all $6.2 \times 10^{-5}$. The SNRs were 14.06 dB, 14.52 dB, 14.98 dB for PCE and CEEs of 5% and 10%, respectively. As such, the received voice signal was clear. Further, the MSEs of the original and the received

voice signals using 16-QAM with a BER of $8.5 \times 10^{-4}$, for PCE and CEEs of 5% and 10%, were $1.5 \times 10^{-4}$, $1.4 \times 10^{-4}$, $1.4 \times 10^{-4}$, respectively. The SNRs were 21.05 dB, 21.54 dB, and 23.46 dB, for PCE and CEEs of 5% and 10%, respectively. Thus, the simulation results showed that the proposed UFMC-based underwater acoustic transmission scheme would be suitable for voice communication.

## 4. Conclusions

In this paper, the UFMC-based underwater transceiver with the integration of a G.729 voice encoder, an LDPC code, adaptive 4-QAM, and 16-QAM, and a power assignment mechanism, was proposed. The BERs and power-saving ratios for the proposed underwater transceiver with a PCE and CEEs of both 5% and 10% were demonstrated. The simulation results showed that the PCE outperformed CEEs of 5% and 10%, under 4-QAM, with gains of 0.5 and 0.9 dB, respectively, with an MSE of $6.2 \times 10^{-5}$. The PCE outperformed CEEs of 5% and 10%, under 16-QAM, with gains of 0.5 and 2.4 dB, respectively, with an MSE of $1.5 \times 10^{-4}$. The received voice signal was clear, and the received transmission BER was below $10^{-3}$. Thus, the UFMC-based underwater transceiver with an LDPC code would be suitable for future underwater voice transmission.

**Funding:** This research received no external funding.

**Acknowledgments:** The author acknowledges the valuable comments of the reviewers.

**Conflicts of Interest:** The author declares no conflict of interest.

## Abbreviations

| | |
|---|---|
| 3GPP-lTE | 3rd generation partnership project long-term evolution |
| 5G | Fifth-generation |
| AIC | Active interference cancellation |
| AWGN | Additive white Gaussian noise |
| BER | Bit error rate |
| BPSK | Binary phase-shift keying |
| CEEs | Channel estimation errors |
| CP | Cyclic prefix |
| CSMA/CA | Carrier sense multiple access with collision avoidance |
| DSL | Digital subscriber line |
| DM | Direct mapping |
| FBMC | Filter bank multicarrier |
| GFDM | Generalized frequency division multiplexing |
| GS | Gold sequence |
| IDFT | Inverse discrete Fourier transform |
| IoT | Internet of Things |
| LDPC | Low-density parity-check |
| M2M | Machine-to-machine |
| MIMO | Multi-input multi-output |
| MSE | Mean square error |
| OFDM | Orthogonal frequency-division multiplexing |
| OQAM | Offset quadrature amplitude modulation |
| OVSF | Orthogonal variable spreading factor |
| PAPR | Peak-to-average power ratio |
| PCE | Perfect channel estimation |
| QAM | Quadrature amplitude modulation |
| *QPSK* | Quadrature phase-shift keying |
| RB | Resource block |
| SNR | Signal-to-noise ratios |
| UAC | Underwater acoustic communication |
| UFMC | universal filtered multi-carrier |

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
