# Peer review of "UFMC-Based Underwater Voice Transmission Scheme with LDPC Codes"

_applsci, doi:10.3390/app11041818_

Round 1

Reviewer 1 Report

The introduction section was long and I had difficulty understanding the point they were trying to make. Other than that, the paper is  well written.  

Author Response

Point 1: The introduction section was long and I had difficulty understanding the point they were trying to make. Other than that, the paper is well written.

The author acknowledges the support of the valuable comments of the reviewer.

Response 1:

The introduction section was long

In introduction section

“Future 5G mobile networks include characteristics for data transmission rates up to 2 Gb/s, network latency up to 1 ms, and the number of serviced devices up to one million per square kilometer. OFDM has certain drawbacks, such as CP overhead, sensitivity to frequency offset, and high peak-to-average power ratio (PAPR).”

has been deleted.

The introduction presented many relevant past studies, and these relevant past studies could be inspired the design idea of underwater transceiver.

I had difficulty understanding the point they were trying to make.

An underwater universal filtered multi-carrier (UFMC)-based voice transceiver design with a LDPC code is an interesting research topic. In the paper, an underwater universal filtered multi-carrier (UFMC)-based voice transceiver with the integration of a G.729 voice encoder, a LDPC code, adaptive4-QAM, and 16-QAM, and a power assignment mechanism, was proposed. The QAM BER performances of the proposed UFMC-based underwater transceiver with a LDPC code, a PCE and CEEs of 5% and 10%, were demonstrated. The transmission power weighting of the UFMC-based underwater transceiver with a LDPC code and a BER 4 x 10^(-4) , for PCE and CEEs of 5% and 10%, were demonstrated.  The power saving ratio of the UFMC-based underwater transceiver with a LDPC code and a BER of 4 x 10^(-4) , for PCE and CEEs of 5% and 10%, were demonstrated. The received voice signal using 16-QAM with a LDPC code , a BER of 8.5x10^(-4) , and a MSE of 0.000015.              for a CEE of 5%, were also demonstrated.

UFMC modulation with the (2000, 1000) LDPC code was an advanced UAC scheme, and high spectral efficiency, high transmission data rate, low transmission BER, and real time voice transmission could be achieved. The underwater channel bandwidth was 20 KHz, and 512-point inverse discrete Fourier transform (IDFT), utilizing 10 sub-bands with 20 subcarriers each. G.729 encoder could compress voice signal, and voice signal transmission bits were reduced. The quality of service of BER for underwater voice signal transmission was below 10^(-3). The adaptive 4-QAM oand 16-QAM modulations with the integration of 1/30, 2/30, …, 30/30 power levels mechanism, 60 transmission mode was achieved to combat diversification underwater channel fading. The quality of service of BER for underwater voice signal transmission was meted, real time and low power underwater voice signal transmissions were also achieved.

Other than that, the paper is well written.

The author acknowledges the support of the valuable comments of the reviewer.

Reviewer 2 Report

In this manuscript, the UFMC-based underwater transceiver with the integration of a G.729 voice encoder, an LDPC code, adaptive 4-QAM, and 16-QAM, and a power assignment mechanism, was proposed. The transceiver in underwater environment will be of interest to many readers. However, UFMC method and LDPC coding have already been studied, respectively. Therefore, it is necessary to differentiate it from the existing methods.

  1. The introduction presents many relevant past studies. But what is the problem you want to solve in the author's manuscript? What is the author's originality in this paper?
  2. Equation (1) : What is X_k in equation (1)? There is only a description of X_ik.
  3. Table 1 : What is the background on which the various parameter values are determined in Table 1? What is the effect of changing these values? It is necessary to observe the performance change according to the various parameter values.
  4. Equation (2) : What is V_1, V_2,...V_B in equation (2)? There is only a description of V_ik.
  5. line 202, 205 : What does delta mean in line 202 and 205? And how was this value set in the simulation?
  6. Please show the channel transmission characteristics (sound speed profile, channel delay profile, Doppler spectrum etc.) applied to the simulation.
  7. In real world, Doppler spreading effect also affects performance. What is the impact on this?
  8. It is difficult to distinguish whether the bit error rate results presented in the paper are based on the UFMC method or LDPC channel coding. If the author wants to analyze the UFMC method, it is necessary to show the performance separately from LDPC.
  9. If possible, I hope that the experimental results are presented.

Author Response

The author acknowledges the support of the valuable comments of the reviewer.

Point 1: The introduction presents many relevant past studies. But whatis the problem you want to solve in the author's manuscript?What is the author's originality in this paper?

Point 8: It is difficult to distinguish whether the bit error rate results presented in the paper are based on the UFMC method orLDPC channel coding. If the author wants to analyze the UFMC method, it is necessary to show the performance separatelyfrom LDPC.

Point 9: If possible, I hope that the experimental results are presented.

An underwater universal filtered multi-carrier (UFMC)-based voice transceiver design with a LDPC code was an interesting research topic. In the paper, an underwater universal filtered multi-carrier (UFMC)-based voice transceiver with the integration of a G.729 voice encoder, a LDPC code, adaptive4-QAM, and 16-QAM, and a power assignment mechanism, was proposed. The QAM BER performances of the proposed UFMC-based underwater transceiver with a LDPC code, a PCE and CEEs of 5% and 10%, were demonstraed. The transmission power weighting of the UFMC-based underwater transceiver with n LDPC code and a BER 4x10^(-4) , for PCE and CEEs of 5% and 10%, were demonstraed.  The power saving ratio of the UFMC-based underwater transceiver with n LDPC code and a BER of 4x10^(-4) , for PCE and CEEs of 5% and 10%, were demonstraed. The received voice signal using 16-QAM with a LDPC code , a BER of 8.5x10^(-4) , and a MSE of 0.000015.              for a CEE of 5%, were also demonstraed.

Point 2: Equation (1) : What is X_k in equation (1)? There is only a description of X_ik.

 are the  complex adaptive QAM symbols for the ith sub-band, and the kth user,

has been revised to

 is a vector of the complex adaptive QAM symbols for the ith sub-band, and the kth user,

Point 3: Table 1 : What is the background on which the various parameter values are determined in Table 1? What is the effect of changing these values? It is necessary to observe the performance change according to the various parameter values.

UFMC modulation with (2000, 1000) LDPC code was an advanced UAC scheme, and high spectral efficiency, high transmission data rate, low transmission BER, and real time voice transmission could be achieved. The underwater channel bandwidth was 20 KHz, and 512-point inverse discrete Fourier transform (IDFT), utilizing 10 sub-bands with 20 subcarriers each. G.729 encoder could compress voice signal, and voice signal transmission bits were reduced. The quality of service of BER for underwater voice signal transmission was below 10^(-3). The adaptive 4-QAM oand 16-QAM modulations with the integration of 1/30, 2/30, …, 30/30 power levels mechanism, 60 transmission mode is achieved to combat diversification underwater channel fading. The quality of service of BER for underwater voice signal transmission was meted, real time, and low power underwater voice signal transmission were also achieved.

Point 4: Equation (2) : What is V_1, V_2,...V_B in equation (2)? There isonly a description of V_ik.

has been revised to

Point 5: line 202, 205 : What does delta mean in line 202 and 205? And how was this value set in the simulation?

delta was .  delta was increase or decrease transmission power weighting, for each power assignment loop. Higher transmission power weighting could contend with higher channel fading, and lower transmission power weighting could contend with lower channel fading.

Point 6: Please show the channel transmission characteristics (sound speed profile, channel delay profile, Doppler spectrum etc.) applied to the simulation.

Point 7: In real world, Doppler spreading effect also affects performance. What is the impact on this?

The doppler spreading effect affected the frequency offset of the received signals, and the BER performance of the transceiver was degraded. In the underwater channel model, the water sound speed and the doppler spread were 1539 m/s, and 400 Hz, respectively. In addition, the channel delay profile was generated by using MATLAB-based underwater channel model [26].

Round 2

Reviewer 2 Report

The author did not respond to all of the reviewers' opinions. However, errors in some equations have been corrected, and the contents of the table have been supplemented. Nevertheless, some important points remain.

  1. A figure of the channel transfer characteristics applied to the simulation should be presented. General underwater acoustic communication simulation provides information on this. It is not clear whether it means that the author used the same channel transfer characteristics as in the reference [26] presented in the manuscript. A channel transmission characteristic as shown in Fig. 4 of Reference [26] is required.

  1. The Doppler spreading effect suggested by the reviewer's opinion means spreading, not Doppler frequency offset. Doppler spread is a measure of spectral broadening caused by the time rate of change of the wireless channel and is defined as the range of frequencies over which the received Doppler spectrum is essentially nonzero. The Doppler spread shown in Table 1 of Reference [26] has the same meaning. Accordingly, the 400 Hz Doppler spread proposed in the revised manuscript seems to be an impractical value.

Author Response

The author acknowledges the support of the valuable comments of the reviewer.

Point 1:

Fig.3 presents the simulated impulse response of time varying shallow underwater channel.

Point 2:

The Doppler spread has been revised to Doppler shift.

Doppler shift (frequency offset) with 400 Hz is a practical value.

The Matlab-based channel model do not provide Doppler spread parameter.

“The doppler spread effect affected the frequency offset of the received signals, and the BER performance of the transceiver was degraded.”

has been deleted.
